# Scaling Probabilistic Circuits via Monarch Matrices

**Honghua Zhang** [* 1]  **Meihua Dang** [* 2]  **Benjie Wang** [* 1]  **Stefano Ermon** [2]  **Nanyun Peng** [1]  **Guy Van den Broeck** [1]

## Abstract

Probabilistic Circuits (PCs) are tractable representations of probability distributions allowing for exact and efficient computation of likelihoods and marginals. Recent advancements have improved the scalability of PCs either by leveraging their sparse properties or through the use of tensorized operations for better hardware utilization. However, no existing method fully exploits both aspects simultaneously. In this paper, we propose a novel sparse and structured parameterization for the sum blocks in PCs. By replacing dense matrices with sparse Monarch matrices, we significantly reduce the memory and computation costs, enabling unprecedented scaling of PCs. From a theory perspective, our construction arises naturally from circuit multiplication; from a practical perspective, compared to previous efforts on scaling up tractable probabilistic models, our approach not only achieves state-of-the-art generative modeling performance on challenging benchmarks like Text8, LM1B and ImageNet, but also demonstrates superior scaling behavior, achieving the same performance with substantially less compute as measured by the number of floating-point operations (FLOPs) during training.

## 1. Introduction

Probabilistic circuits (PCs) are a unifying representation of tractable probability distributions through computation graphs (Choi et al., 2020; Darwiche, 2003). The key property that separates PCs from other deep generative models such as flow-based models (Papamakarios et al., 2021) and VAEs (Kingma & Welling, 2013) is their *tractability*. This property enables PCs to compute various queries, including marginal probabilities, exactly and efficiently (Vergari et al.,

2021). The tractability of PCs have been exploited in a number of domains, including fair and explainable machine learning (Choi et al., 2021; Wang et al., 2021b), causal inference (Wang et al., 2021a; Zečević et al., 2021; Wang et al., 2022; Wang & Kwiatkowska, 2023), controllable generation (Liu et al., 2024b; Zhang et al., 2024), and neuro-symbolic AI (Ahmed et al., 2022; Maene et al., 2024).

Recent advancements in PC learning techniques (Liu et al., 2023b; Gala et al., 2024) and efficient tensorized implementations (Peharz et al., 2020; Dang et al., 2021; Liu et al., 2024a) have significantly enhanced the expressiveness and scalability of PCs. However, to further boost the performance of PCs, simply scaling up the model size is insufficient; we need to better utilize the available capacity. To this end, Dang et al. (2022) found that learned PCs empirically exhibit significant sparsity, and leveraged this observation to iteratively learn PC structures through pruning and growing. However, the sparse connections learned are arbitrary, making them difficult to tensorize and parallelize effectively.

In this paper, we focus on leveraging *structured* sparse parameterizations for the sum blocks in PCs, which represent linear maps. To the best of our knowledge, all previous circuit architectures utilizing tensorized operations have relied on dense matrices to parameterize these linear maps (Peharz et al., 2020), which incur a quadratic cost in the number of nodes. Inspired by recent advances in low-rank approximations for transformers (Hu et al., 2022), we propose a novel, more efficient parameterization for the sum blocks.

We begin by illustrating how our parameterization naturally arises from *circuit multiplication*. Previous analysis (Shen et al., 2016; Vergari et al., 2021) showed that multiplying two (compatible) circuits could result in a quadratic increase in size in the worst case. However, we observe that the linear maps in the sum blocks generated by multiplication are not dense but rather the *Kronecker product* of the linear maps from the original blocks, which can be implemented more efficiently. Further, by explicitly materializing this map as interleaving sums and permutations, we identify an interesting connection between these product circuits and some recently introduced class of structured matrices including *Butterfly* matrices (Dao et al., 2019) and *Monarch* matrices (Dao et al., 2022). Building on this insight, we propose replacing the dense linear maps in PCs with Monarch layers.

---

[*]Equal contribution  [1]Department of Computer Science, University of California, Los Angeles [2]Department of Computer Science, Stanford University. Correspondence to: Honghua Zhang <hzhang19cs@gmail.com>.

*Proceedings of the 42$^{nd}$ International Conference on Machine Learning*, Vancouver, Canada. PMLR 267, 2025. Copyright 2025 by the author(s).

In our empirical evaluation[1], we demonstrate that by replacing dense matrices in PCs with structured Monarch matrices, we are able to scale PCs to orders-of-magnitude larger hidden sizes and, among a variety of tractable generative models, we are able to achieve state-of-the-art density estimation performance on various benchmarks, including ImageNet32/64 (Deng et al., 2009) for image modeling and Text8 (Mahoney, 2011) and LM1B (Chelba et al., 2013) for language modeling. Furthermore, we show that, compared to circuits with dense layers, the ones with Monarch layers yield significantly better scaling curves: they can achieve the same performance with substantially fewer floating-point operations (FLOPs) in training and inference.

## 2. Tensorized Probabilistic Circuits

**Notation**  We use uppercase to denote variables (e.g. $X$) and lowercase to denote values of variables (e.g. $x$). We use boldface to denote sets of variables/values (e.g. $\boldsymbol{X}, \boldsymbol{x}$).

**Definition 2.1** (Probabilistic Circuit). A PC $\mathcal{C} = (\mathcal{G}, \boldsymbol{\theta})$ represents a joint probability distribution over random variables $\boldsymbol{X}$ through a directed acyclic (computation) graph (DAG) $\mathcal{G}$ parameterized by $\boldsymbol{\theta}$. Specifically, the DAG $\mathcal{G}$ consists of three types of nodes – *sum*, *product*, and *leaf* nodes. Each leaf node $n$ is associated with a non-negative function $f_n(X_n)$ over some variable $X_n$, called its *scope* $\boldsymbol{X}_n := \{X_n\}$. The scope of any sum or product node $n$ is defined to be $\boldsymbol{X}_n := \bigcup_{c \in \text{ch}(n)} \boldsymbol{X}_c$, where $\text{ch}(n)$ denotes the children of $n$ in $\mathcal{G}$. Each node $n$ represents a probability distribution $p_n$ over its scope $\boldsymbol{X}_n$, defined recursively by:

$$p_n(\boldsymbol{X}_n) = \begin{cases} f_n(X_n) & \text{if } n \text{ is a leaf node} \\ \prod_{c \in \text{ch}(n)} p_c(\boldsymbol{X}_c) & \text{if } n \text{ is a product node} \\ \sum_{c \in \text{ch}(n)} \theta_{c|n} \cdot p_c(\boldsymbol{X}_c) & \text{if } n \text{ is a sum node} \end{cases}$$

where for each leaf node, function $f_n(X_n)$ represents a normalized univariate probability mass/density function (e.g. Categorical, Gaussian); and for every sum node $n$, $\theta_{c|n}$ is a non-negative weight associated with the edge $(n, c)$ in the DAG. If $\sum_{c \in \text{ch}(n)} \theta_{c|n} = 1$, then the PC computes a normalized joint probability mass/density function. The function represented by a PC, denoted $p_{\mathcal{C}}(\boldsymbol{X})$, is the function represented by its root node; and the size of a PC, denoted $|\mathcal{C}|$, is the number of edges in its graph.

It is immediate from the definition that one can evaluate a PC's function with a single traversal through its computation graph. The distinguishing feature of PCs compared to other computation graphs such as neural networks is that one can also efficiently compute *marginals* under the following restrictions on the node scopes:

---

[1] Code available at https://github.com/wangben88/MonarchCircuits

**Definition 2.2** (Smoothness and Decomposability). A sum node is *smooth* if all of its children have the same scope. A product node is *decomposable* if its children have disjoint scope. A PC is smooth (resp. decomposable) if all of its sum (resp. product) nodes are smooth (resp. decomposable).

In practice, probabilistic circuit graphs are typically designed in a tensorized manner, in which sets of nodes of the same type (sum, product, leaf) and with the same scope are grouped together as a *block*; the computation graph is then specified through connections between the blocks (Peharz et al., 2020; Liu et al., 2024a; Loconte et al., 2024a). We write $\boldsymbol{n}$ to denote a node block and $|\boldsymbol{n}|$ for the number of nodes in the block.

**Definition 2.3** (Sum Block). A sum block $\boldsymbol{n}$ has a set of child blocks $\{\boldsymbol{c}^{(i)}\}_{i=1}^m$, such that each sum node in the block is connected to every node in each of the child blocks. We can write $W \in \mathbb{R}^{|\boldsymbol{n}| \times (\sum_{i=1}^m |\boldsymbol{c}^{(i)}|)}$ for the weight matrix.

**Definition 2.4** (Product Block). A product block $\boldsymbol{n}$ has a set of child blocks $\{\boldsymbol{c}^{(i)}\}_{i=1}^m$. We define two types of product node block with different connectivity:

- Hadamard $\bigodot$: If $|\boldsymbol{c}^{(i)}| = |\boldsymbol{n}|$ for all $i = 1, \ldots, n$, then we define a Hadamard product block where $\boldsymbol{n} = \bigodot_{i=1}^m \boldsymbol{c}^{(i)}$.

- Kronecker $\bigotimes$: If $|\boldsymbol{n}| = \prod_{i=1}^m |\boldsymbol{c}^{(i)}|$, then we can define a Kronecker product node block where $\boldsymbol{n} = \bigotimes_{i=1}^m \boldsymbol{c}^{(i)}$.

Sum blocks represent parameterized linear maps, while product blocks represent fixed *multi*linear maps. Typically, for smooth and decomposable PCs, one builds the circuit by alternating between sum and product blocks (i.e., children of sum blocks are product blocks, children of product blocks are sum/leaf blocks). In this paper, we focus on the parameterization of the sum blocks, which is independent from the choice of Hadamard or Kronecker product blocks (we use Hadamard product blocks for our experiments).

**Measuring the sizes of PCs**  The size of a probabilistic circuit (number of edges) determines the number of FLOPS needed for a forward pass. This can be roughly characterized as a function of two parameters: the number of sum blocks $n$, and the input/output dimension of the largest sum block, which we call the *hidden size* $h$. The size of the circuit is then bounded by $O(nh^2)$.

## 3. From Circuit Multiplication to Generalized Monarch Matrices

In this section, we describe from first principles our construction of PCs parameterized by (generalized) Monarch matrices. Firstly, in Section 3.1, we focus on the fundamental operation of *circuit multiplication*, and show how

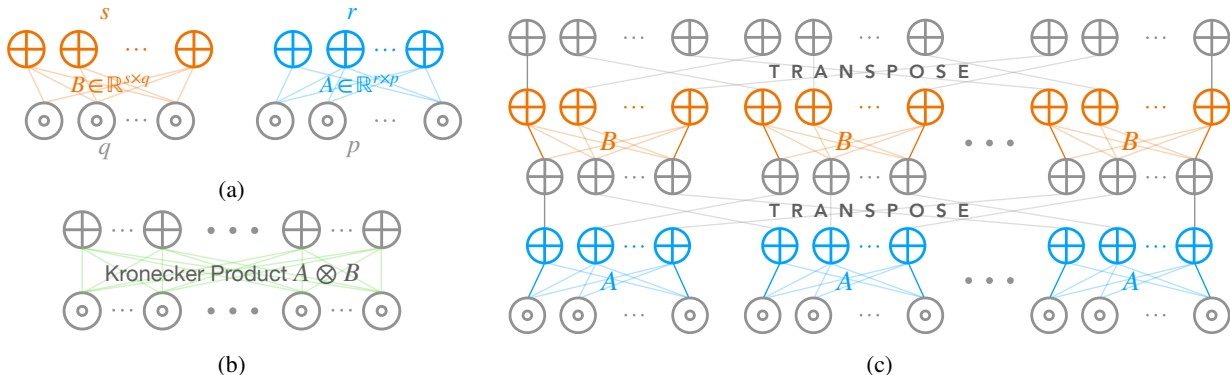

Figure 1: **Probabilistic circuits architecture illustration with Monarch matrices.** (a) Two sum blocks in PCs with weight matrices $A, B$ of arbitrary dimensions. (b) A constructed sum block with weight matrix as the Kronecker product $A \otimes B$, representing the circuit product of two sum blocks. (c) Efficient circuit representation for the linear transformation $A \otimes B$.

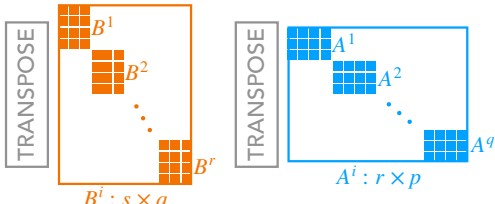

Figure 2: Generalized two-layer Monarch matrix.

the resulting circuit exhibits structured sparsity, leading to PCs parameterized by Monarch matrices. In Section 3.2, we extend this analysis to the multiplication of more than two PCs, and derive a new *multi-layer* Monarch matrix parameterization. Finally, in Section 3.3, for a PC with hidden size $h$, we show how this can be used to control the compute and memory used between $O(h \log(h))$ and $O(h^2)$, which effectively interpolates between the Monarch and the sparser *Butterfly matrices*.

### 3.1. Materializing Circuit Multiplication

Given two circuits $\mathcal{A}$ and $\mathcal{B}$, the goal of circuit multiplication is to construct a tractable (i.e. smooth and decomposable) circuit $\mathcal{C}$ such that $p_{\mathcal{C}}(\boldsymbol{x}) \propto p_{\mathcal{A}}(\boldsymbol{x}) \cdot p_{\mathcal{B}}(\boldsymbol{x})$. If $\mathcal{A}$ and $\mathcal{B}$ are structured-decomposable with respect to the same vtree, i.e., the product nodes in $\mathcal{A}$ and $\mathcal{B}$ always factor the same way, then $\mathcal{C}$ can be constructed with at most a quadratic increment in size (Shen et al., 2016; Vergari et al., 2021). The corresponding algorithm essentially constructs a new node for each pair of nodes with the same scope.

Here we focus on the local operation of multiplying two sum blocks. Given two sum blocks with weight matrices $A \in \mathbb{R}^{r \times p}$ and $B \in \mathbb{R}^{s \times q}$ (Figure 1a), their product can be represented as a sum block with input dimension $pq$ and output dimension $rs$, and weight matrix given by the

Kronecker product $A \otimes B$ (Vergari et al., 2021). Figure 1b shows a circuit materialization of $A \otimes B$, consisting of $O(rspq)$ edges. However, the linear transformation given by $A \otimes B$ can actually be executed in a significantly more efficient way: letting $\boldsymbol{x} \in \mathbb{R}^{p \times q}$ be an input tensor, we compute the linear transformation $(A \otimes B)\boldsymbol{x}$ as

$$
\begin{aligned}
&((A \otimes B)\boldsymbol{x})_{kl} \\
&= \sum_{ij}(A \otimes B)_{kl,ij}\boldsymbol{x}_{ij} = \sum_{i,j}A_{ki}B_{lj}\boldsymbol{x}_{ij} \\
&= \sum_{j}B_{lj}\sum_{i}A_{ki}\boldsymbol{x}_{ij} = \sum_{j}B_{lj}(A\boldsymbol{x})_{kj} \\
&= \sum_{j}B_{lj}(A\boldsymbol{x})_{jk}^{T} = (B(A\boldsymbol{x})^{T})_{lk} = (B(A\boldsymbol{x})^{T})_{kl}^{T};
\end{aligned}
$$

hence, we have

$$
(A \otimes B)\boldsymbol{x} = (B(A\boldsymbol{x})^{T})^{T}. \tag{1}
$$

We attempt to materialize Equation 1 as a circuit, as shown in Figure 1c, with $\boldsymbol{x} \in \mathbb{R}^{p \times q}$ viewed as a flattened 1-d tensor of dimension $pq$. The total number of edges is bounded by $O(rpq + srq)$. For a rough comparison against the naive construction of $A \otimes B$, if $A$ and $B$ are both of dimension $m \times m$, then the naive circuit construction contains $O(m^4)$ edges while the construction based on Equation 1 contains only $O(m^3)$ edges.[2]

Although the circuit shown in Figure 1c is constructed by multiplying two sum blocks, it can also be interpreted as

---

[2]The code for circuit multiplication in the official implementations of Wang & Kwiatkowska (2023) and Loconte et al. (2024b) also achieve this complexity through einsum operations; though this is not explicitly described in either of the papers, which report the looser bound of $O(m^4)$, i.e., square of the circuit size. Crucially, we show that the product can be explicitly materialized as a circuit, which was missed by these prior works. This allows us to directly apply standard PC inference and learning algorithms to the product circuit, such as parameter learning with expectation-maximization.

a *sparse* representation for some linear transformation $\mathcal{M}$ from $\mathbb{R}^{pq}$ to $\mathbb{R}^{rs}$. Furthermore, in this interpretation, each of the $A$ blocks (and each of the $B$ blocks) *do not need to have the same parameters* for $\mathcal{M}$ to be valid. Hence, we "untie" the parameters of $A$ and $B$ to obtain the (generalized) Monarch transformation $\mathcal{M}$.

**Definition 3.1** (Generalized Monarch Matrices I). Given $A \in \mathbb{R}^{r \times p \times q}$ and $B \in \mathbb{R}^{s \times q \times r}$, we define $\mathcal{M}$ to be the linear transformation from $\mathbb{R}^{pq}$ to $\mathbb{R}^{rs}$ such that

$$(\mathcal{M}x)_{kl} = \sum_{i,j} A_{kij}B_{ljk}\boldsymbol{x}_{ij} = (B * (A * \boldsymbol{x})^T)^T;$$

where $A * \boldsymbol{x}$ denotes a "batched" linear transformation

$$(A * \boldsymbol{x})_{kj} = \sum_i A_{kij}\boldsymbol{x}_{ij}$$

with $j$ enumerating through the "batch" dimension.

We visualize the generalized Monarch matrix in Figure 2. Here, the 2D matrices $A^j$ and $B^i$ are actually slices of the 3D tensors $A$, $B$ in Definition 3.1; specifically $(A^j)_{ki} = A_{kij}$ and $(B^k)_{lj} = B_{ljk}$.

Thus, to enable structured sparsity in PCs, we propose to replace the dense matrices in sum blocks with generalized Monarch matrices, parameterized by the tensors $A, B$. In the case where we have $h$ input and output nodes (with $p = q = r = s = \sqrt{h}$), then the compute and memory requirements are $O(h^{3/2})$ as compared to the $O(h^2)$ cost for a dense PC. The interpretation as circuit multiplication gives us a principled means to initialize the parameters of such a circuit: namely, train two smaller PCs $\mathcal{A}, \mathcal{B}$ with *dense* layers and hidden sizes $\sqrt{h}$ and then multiply them to obtain a circuit representing the distribution $p_{\mathcal{C}}(\boldsymbol{x}) \propto p_{\mathcal{A}}(\boldsymbol{x})p_{\mathcal{B}}(\boldsymbol{x})$. We can then untie the parameters of $\mathcal{C}$ during training to leverage the fully general Monarch paramterization.

### 3.2. Multiplying Multiple PCs

A natural way to further generalize the construction above would be to consider products of *multiple* PCs, or more specifically, Kronecker products of *multiple* matrices. We generalize Equation 1 as:

$$\begin{aligned}(A^1 \otimes A^2 &\cdots \otimes A^d)\boldsymbol{x} \\ &= (A^d \ldots (A^2(A^1\boldsymbol{x})^S)^S \ldots )^S;\end{aligned} \quad (2)$$

with $A^t \in \mathbb{R}^{m_t \times n_t}$ and $x \in \mathbb{R}^{n_1 \times \cdots \times n_d}$; the superscript $S$ denotes the *left shifting operation* where, e.g., $x^S{}_{jki} = x_{ijk}$. Similarly, we can materialize Equation 2 as a circuit, untie the shared parameters among the blocks $A^1, \ldots, A^d$, and then obtain the (further) generalized construction of Monarch matrices.

**Definition 3.2** (Generalized Monarch Matrices II). Let $\{A^t\}_{1 \leq t \leq d}$ be $d$ tensors, where $A^t$ has dimensions $m_t \times$ $n_t \times n_{t+1} \times \cdots \times n_d \times m_1 \times \cdots \times m_{t-1}$. We define the generalized Monarch matrix $\mathcal{M} \in \mathbb{R}^{(m_1 \times \cdots m_d) \times (n_1 \times \cdots n_d)}$

$$\begin{aligned}(\mathcal{M}&x)_{j_1,j_2,\ldots,j_d} \\ &= \sum_{i_1,i_2,\ldots i_d} \left(\prod_{1 \leq t \leq d} A^t_{j_t i_t i_{t+1}\ldots i_d j_1 \ldots j_{t-1}}\right) x_{i_1 \ldots i_d} \\ &= (A^d * (A^{d-1} * (\cdots * (A^2 * (A^1 * x)^S)^S \ldots )^S.\end{aligned}$$

Here $x^t := (A^t * \cdots * (A^1 * x)^S \cdots )^S$ is of dimension $n_{t+1} \times \cdots n_d \times m_1 \times \cdots m_t$ and $A^{t+1} * x^t$ denotes a batched linear transformation such that

$$(A^t * x^{t-1})... = \sum_{i_t} A^t_{j_t i_t \cdots i_d j_1 \cdots j_{t-1}} x_{i_t \cdots i_d j_1 \cdots j_{t-1}},$$

with $i_{t+1} \cdots i_d j_1 \cdots j_{t-1}$ enumerating over the batch dimension. Note that the circuit materialization of this construction of Monarch matrix consists of $d$ consecutive sum blocks, so we call it a *d-layer Monarch matrix*.

Despite the complexity of this definition, to construct a PC $\mathcal{A}$ with Monarch layers of hidden size $h = \prod_{1 \leq t \leq d} h_t$, we really just need to multiply $d$ PCs $\mathcal{A}_t$, each with dense layers of hidden size $h_t$. As in the case of two circuits, we can first train the smaller PCs $\mathcal{A}_t$ with dense layers and use their parameters as an initialization point for the training of $\mathcal{A}$ where $p_{\mathcal{A}}(\boldsymbol{x}) \propto \prod_t p_{\mathcal{A}_t}(\boldsymbol{x})$. Such parameter initialization significantly improves the result of training PCs with Monarch layers of large hidden sizes, and we refer readers to Section 5.2 for details.

### 3.3. Interpolating Butterfly and Monarch Matrices

To conclude this section, we draw an interesting connection between the generalized $d$-layer Monarch matrices and the *butterfly matrices* (Parker, 1995; Dao et al., 2019; Meng et al., 2022). Butterfly matrices are a class of expressive structured matrices, constructed as the product of sparse matrices known as butterfly factor matrices. For ease of exposition, we will describe here butterfly factor matrices of size $D \times D$, where $D$ is a power of 2.

**Definition 3.3** (Butterfly Factor). Given any $D = 2^d$ and $1 \leq i \leq d$, a butterfly factor matrix $B(i, D)$ is a sparse matrix with the following sparsity pattern:

- $B(1, D)$ has non-zero elements only along the diagonals of the four $\frac{D}{2} \times \frac{D}{2}$ submatrices.

- $B(i, D)$ is a block-diagonal matrix with block size $\frac{D}{2^{i-1}} \times \frac{D}{2^{i-1}}$, where each block is a $B\left(1, \frac{D}{2^{i-1}}\right)$ butterfly factor.

Examples of butterfly factors for $D = 16$ are shown in Figure 3. A $D \times D$ *butterfly matrix* is given by the matrix multiplication $B(D) := B(1, D)B(2, D) \ldots B(d, D)$.

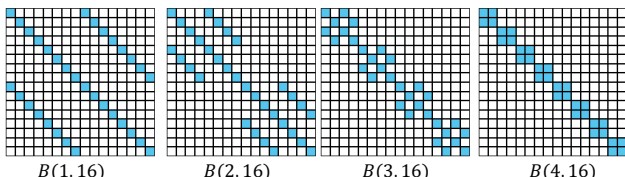

$B(1,16)$     $B(2,16)$     $B(3,16)$     $B(4,16)$

Figure 3: Illustration of Butterfly factors (Meng et al., 2022).

Even though this is a product of *sparse matrices*, we show that in fact it can be interpreted as a product of *dense tensors*, specifically, generalized Monarch matrices:

**Theorem 3.4.** *Suppose that $m_t = n_t = 2$ for all $1 \leq t \leq d$. Then the generalized $d$-layer Monarch matrix $\mathcal{M}$ is a butterfly matrix of dimension $2^d \times 2^d$.*

More generally, suppose that we want to construct a generalized Monarch matrix of dimension $h \times h$. We can interpolate between butterfly (consisting of $\log_2(h)$ different $(\log_2(h)+1)$-dimensional tensors of shape $2 \times 2 \times ... \times 2$) and Monarch matrices (consisting of 2 different 3-dimensional tensors of shape $h^{1/2} \times h^{1/2} \times h^{1/2}$) as follows. We can first pick a number $c$ such that $c^d = h$ for some $d$, and then the construction in Defintion 3.2 would give us a $\log_c(h)$-layer Monarch matrix with each layer (in terms of the circuit materialization) consisting of $hc$ edges; i.e., the # of FLOPs needed for the linear transformation is given by $hc \log_c(h)$. We have the butterfly matrix if $c = 2$ and the commonly known Monarch matrix (Dao et al., 2022) if $c = \sqrt{h}$. If $h$ is nice enough, e.g., $2^{20}$, we can freely choose $c$ to interpolate between 2 and $\sqrt{h}$, which corresponds to $d$-layer Monarch matrices with $d$ ranging from $\log_2(h)$ to 2, each having different degrees of sparsity.

## 4. Scaling PCs via Monarch Matrices

Employing structured matrices improves two aspects of probabilistic circuits. First, fixing the hidden size, the PCs require fewer FLOPs so training will be more efficient. Second, it reduces the memory consumption, so models with larger hidden sizes are able to fit in the GPU memory. These two benefits make it possible to scale up probabilistic circuits to much larger hidden sizes. In this section, we discuss some of the practical choices to be made for learning PCs with (generalized) Monarch parameterizations, the effect of which we investigate empirically through ablation studies.

**Choices of Monarch Structures** As described at the end of Section 3, given a desired linear transformation from $\mathbb{R}^h$ to $\mathbb{R}^h$, there are different choices of Monarch structures. For example, $h = 2^{18}$ can be factorized as $2^9 \times 2^9$, $2^6 \times 2^6 \times 2^6$, $(2^3)^6$ etc., where $2^9 \times 2^9$ gives the standard two-layer Monarch matrix and $2^6 \times 2^6 \times 2^6$ gives a three-layer Monarch matrix. For hidden sizes like $2^{19}$ that cannot

be nicely "factored" into squared matrices, we resort to the almost-squared factorizations like $2^{19} = 2^9 \times 2^{10}$. In Section 5.1 and 5.2, we conduct an empirical study on the scaling behavior of Monarch matrices with different number of layers and show that even though Monarch matrices with more layers exhibit slightly better scaling behavior compared to two-layer Monarch matrices, they induce higher memory consumption and thus are less desirable in practice.

**Initialization** The probabilistic interpretation of Monarch layers as circuit multiplication further provides an intuitive way for initializing large circuits. For instance, to initialize a 2-layer Monarch with hidden size of $h = 2^{20}$, we can multiply two smaller models with hidden size of $h = 2^{10}$ (or any possible factorizations). The smaller models are trained separately to fit the same data distribution and then combined to initialize the larger model's parameters.

**Training** We use a stochastic mini-batch version of Expectation-Maximization optimization (Peharz et al., 2016). Empirically, this approach converges faster and is better regularized compared to EM on the whole dataset.

**Research Questions** In the following experiments, we seek to verify the following hypotheses:

- *Scaling law for Monarch PCs.* Sparse structures achieve better scaling behavior compared to dense ones in terms of bits-per-dimension (BPD) versus floating point operations (FLOPs).

- *Effect of Circuit Multiplication.* Initializations using circuit multiplication leads to better performance.

- *Increasing model size induces sparsity.* For a fixed hidden size, denser matrices perform better, but the performance gap between sparse and dense matrices diminishes quickly as the hidden size increases.

## 5. Experiments

We evaluate our method using generative modeling benchmarks for both text and image data. We use log-likelihoods as a measurement of a model's performance and the number of floating point operations (FLOPs) per dimension as a measurement of a model's efficiency. Given hidden size of $h$, the FLOPs per token is $h^2$ for an HMM and $2h^{3/2}$ for a two layer Monarch-HMM. Details are in Appendix B.

### 5.1. Character-level Language Modeling

**Dataset** Text8 (Mahoney, 2011) is a character-level language modeling dataset with a vocabulary of 27 tokens: the letters 'a'-'z' and the whitespace token. We follow the standard practice of training and evaluating text8 in chunks of length 256 without preprocessing (Hoogeboom et al., 2021).

| Type | Model | BPC (↓) | Time (s) (↓) |
|------|-------|---------|--------------|
| Flow | IAF/SCF | 1.88 | 0.04 |
| Flow | Argmax Coup Flow | 1.80 | 0.40 |
| Diffusion | D3PM Uniform | ≤ 1.61 | 3.60 |
| Diffusion | SEDD Uniform | ≤ 1.47 | - |
| PC | SparsePC | 2.60 | - |
| PC | NPC$^2$ | 3.17 | - |
| PC | HMM | 1.69 | 0.006 |
| PC | Monarch-HMM | **1.57** | 0.017 |

Table 1: **Averaged test set BPC on text8.** Sample times are for generating an example of length 256. Our method outperforms PC baselines, and significantly close the gap between PCs and the other less tractable generative models.

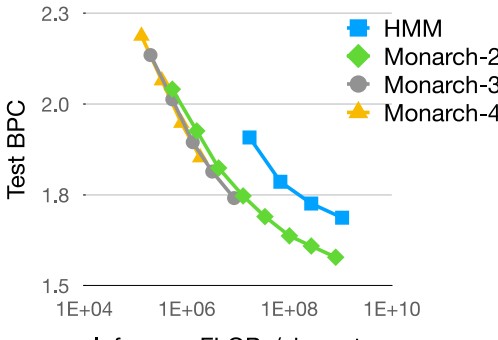

Figure 4: **Scaling curves comparing HMM and varying Monarch structures**. BPC (↓) as a function of training FLOPs per character. Monarch-HMM demonstrates greater efficiency than HMM across varying computational budgets. Sparser Monarch further leads to better scaling behaviors.

| Hidden Size | $2^{12}$ | $2^{13}$ | $2^{14}$ | $2^{15}$ |
|-------------|----------|----------|----------|----------|
| Random Initialization | 2.25 | 2.10 | 1.98 | 1.88 |
| PC Multiplication | 2.14 | 2.01 | 1.90 | 1.81 |

Table 2: Test set BPC (↓) of a 3-layer Monarch-HMM for different hidden sizes comparing random initialization and initialization from PC multiplication with 3 dense HMMs.

**Baselines** We compare our method with probabilistic circuit structures, including SparsePC (Dang et al., 2022), NPC$^2$ (Loconte et al., 2024b), and HMM (Zhang et al., 2023; 2024), as well as other types of less tractable generative models as a reference, such as flow-based models (IAF/SCF (Ziegler & Rush, 2019), Argmax Coupling Flow (Hoogeboom et al., 2021)) and diffusion models (D3PM (Austin et al., 2021), SEDD (Lou et al., 2024)) with uniform transition matrices. The results for SparsePC and NPC$^2$ are obtained by rerunning their official implementations. SparsePC constructs arbitrary sparse structures by iteratively pruning and growing an initial model, which prevents it from scaling to very large hidden size due to system limitations. For HMM, we use dense transition matrices with the largest possible hidden state and run it using our codebase to ensure optimal performance. The results for other generative models are taken from (Austin et al., 2021) and (Lou et al., 2024).

**Benchmark** We report log-likelihood results in bits-per-character (BPC) in Table 1. The results show that Monarch-HMM outperforms all PC models and significantly narrows the gap between PC models and other less tractable generative models. We improve upon the baseline HMM by replacing dense matrices with structured sparse (Monarch) matrices; thus, we are able to scale up hidden size from $2^{15}$ to $2^{19}$. Additionally, Monarch-HMM achieves significantly faster inference, 200x times faster than diffusion models.

**Scaling laws of structured matrices** We also include a plot in Figure 4 comparing HMM and Monarch-HMM in terms of BPC as a function of inference FLOPs. Monarch-HMM is trained with hidden size ranging from $2^{12}$ to $2^{19}$. Monarch-HMM demonstrates greater efficiency than HMM across different computational budgets consistently. We additionally include Monarch-3 and Monarch-4, which represent Monarch-HMM with three and four Monarch layers respectively (Section 3). Since models with more layers have more sparse structures, this suggests that increased

sparsity leads to better scaling behavior. However, sparser structures are constrained by memory consumption (Section 5.2), so we use Monarch-2 as our main result in Table 1.

### 5.2. Ablation and Analysis

We investigate the relationship between hidden size, FLOPs and memory consumption for varying structures, including dense HMM and differernt layers of Monarch-HMM. From Section 5.1, we conclude that sparser structures achieve better FLOPs. In this section, we aim to verify the following hypotheses: (1) Initializations from circuit multiplications give better performance. (2) For a fixed hidden size, denser matrices outperform sparse ones but the performance gap dimish quickly as hidden size increases; and (3) Monarch-HMMs with more layers requires significantly higher memory consumption.

**Initialization matters** The probabilistic interpretation of Monarch matrices as circuit multiplication, as introduced in Section 3, provides an effective approach for initializing the parameters of PCs with Monarch layers. We compare the performance of Monarch-HMM trained with two parameter-initialization strategies: random initialization vs. initialization from multiplying dense HMMs. As shown in Table 2, circuit multiplication as initialization leads to consistent improvement across all scales.

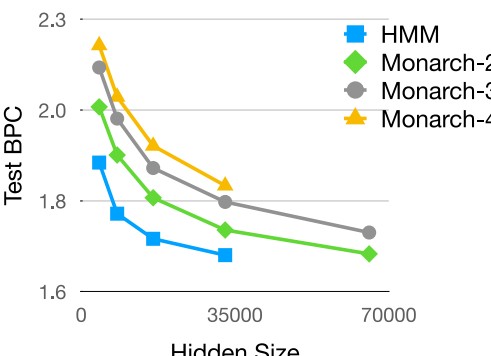

Figure 5: Test set BPC as a function of hidden size for HMM and varying Monarch structures.

| Model | FLOPs $\mathcal{O}(\cdot)$ | Mem Complexity $\mathcal{O}(\cdot)$ | GPU Mem GB |
|---|---|---|---|
| HMM | $h^2$ | $nhB+h^2$ | 288 |
| Monarch 2 | $2h^{3/2}$ | $2nhB+2h^{3/2}$ | 65 |
| Monarch 3 | $3h^{4/3}$ | $3nhB+3h^{4/3}$ | 96 |
| Monarch 4 | $4h^{5/4}$ | $4nhB+4h^{5/4}$ | 128 |

Table 3: **FLOPS and memory consumption for varying structures.** GPU memory is computed when batch size $B=128$, sequence length $n=256$, and hidden size $h=2^{18}$.

**Scaling up hidden size** As shown in Figure 5, bits-per-character improves as hidden size increases. For a fixed hidden size, denser structures always perform better as a dense matrix can always represent a sparse one by incorporating zero values, while having greater capacity in theory. However, this comes at the cost of increased FLOPs as shown in Figure 4. Very interestingly, as the hidden size increases, the performance gap between dense and sparse ones diminishes quickly. This suggests that larger PC models are inherently more sparse, making structured sparse layers increasingly effective in capturing the underlying data distribution as we scale up.

**Sparse structures and memory consumption** For a probabilistic circuit with either dense or sparse matrices, let $n$ be the sequence length, $h$ the hidden size, $d$ the number of Monarch layers, and $B$ the batch size. The training memory consumption consists of: (1) *parameter caching*, which is also linear with FLOPs per character, requiring $\mathcal{O}(h^2)$ for a dense HMM and $\mathcal{O}(dh^{d+1/d})$ for a $d$ layer Monarch); (2) *gradient caching for backward propagation*, which is proportional to the number of nodes (hidden states), requiring $\mathcal{O}(nhB)$ for a dense HMM and $\mathcal{O}(dnhB)$ for a $d$ layer Monarch. This connection is summarized in Table 3. The last column provides an example on GPU memory usage: though Monarch-3/4 have better FLOPs efficiency, training them is impractical due to their memory consumption.

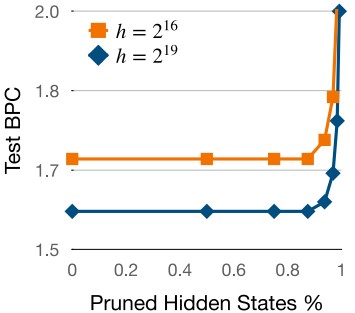

Figure 6: We can effectively prune up to 90% hidden states with no sigificant performance drop for Monarch-HMM.

| Type | Model | Perplexity ($\downarrow$) | Time (s) ($\downarrow$) |
|---|---|---|---|
| Diffusion | D3PM Uniform | $\leq 137.9$ | 1.82 |
| Diffusion | SEDD Uniform | $\leq 40.24$ | - |
| PC | HMM | 320.78 | 0.0026 |
| PC | Monarch-HMM | **190.34** | 0.0095 |

Table 4: **Averaged test set perplexity on LM1B.** Sample times are for generating an example of length 128. Our method outperforms PC baselines, and significantly close the gap between PC and other generative models.

**Sparse hidden representations** Dang et al. (2022) show that learned dense layers in PCs are inherently sparse, motivating our use of Monarch layers. Using Monarch layers, we scale circuits to larger hidden sizes, where memory consumption from caching hidden states (node values) for backward propagation becomes the new bottleneck ($\mathcal{O}(dnhB)$ in Table 3). We investigate whether these representations are also sparse by pruning hidden states during the forward pass of a Monarch-HMM. Figure 6 shows up to 90% of hidden states across all layers can be pruned without a significant drop in log-likelihood.

### 5.3. Token-level Language Modeling

Language modeling for large-scale datasets with large vocabularies using probabilistic circuits has not been previously demonstrated. We present results on the One Billion Word dataset (LM1B)(Chelba et al., 2013) as a proof of concept, demonstrating the scalability of Monarch-HMM. Following D3PM(Austin et al., 2021), all models are trained and evaluated on packed sequences of length 128 using a SentencePiece [3] vocabulary of size 8192. We use HMM as our baseline, as other probabilistic circuit models are unable to scale to this dataset. Additionally, we include results from diffusion models from Austin et al. (2021) and Lou et al. (2024) for reference. Monarch-HMM is trained using the same setup as in the text8 experiments for two epochs.

---

[3] https://github.com/google/sentencepiece

|  | ImageNet 32×32 | | ImageNet 64×64 | |
|  | Lossy | Lossless | Lossy | Lossless |
| --- | --- | --- | --- | --- |
| LVD | 4.39 | - | 4.12 | - |
| LVD-PG | 4.06 | - | 3.80 | - |
| QPC | 4.46 | 5.08 | 4.42 | 5.05 |
| Monarch | **4.01** | **4.62** | **3.74** | **4.33** |
| RealNVP | - | 4.28 | - | 3.98 |
| Glow | - | 4.09 | - | 3.81 |
| VDM | - | 3.72 | - | 3.40 |

Table 5: **Density estimation on image datasets.** Test set log-likelihoods are in bits-per-dimension (lower is better). Our method performs favorably relative to all PC baselines.

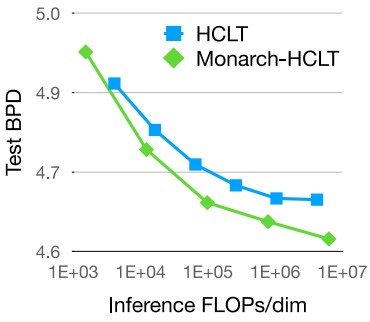

Figure 7: **Results for Training on ImageNet32 Dataset**. Test bits-per-dimension (↓) as a function of training FLOPs per pixel. Monarch-HCLT demonstrates greater efficiency than HCLT as the computation budget increases.

We report test set perplexity and sampling tims in Table 4. While other PC models cannot scale to this range, Monarch-HMM significantly bridges the gap between probabilistic circuits and diffusion models, while retaining the tractability advantage of PCs over diffusion models.

## 5.4. Image Modeling

In this section, we conduct experiments on the ImageNet32 and ImageNet64 datasets, which are downscaled $32 \times 32$ and $64 \times 64$ versions of ImageNet (Deng et al., 2009). To improve modeling performance, and following prior work (Liu et al., 2023a; 2024b;a; Gala et al., 2024), we apply a color transform to the original RGB data, and fit PC models on this transformed data. In particular, we employ a *lossless* YCoCg-R transform (Malvar & Sullivan, 2003) (see Appendix B for details); as such likelihoods on this transformed data are comparable to those on the original RGB dataset. As some prior works on PC learning (Liu et al., 2023a;b) employ a *lossy* quantized YCoCg transform; we also report results using this transform for fair comparison.

To improve training efficiency and enable greater scaling up of hidden size, we split each image into $8 \times 8$ patches and train and evaluate our PCs over these $8 \times 8$ images ($8 \times 8 \times 3 = 192$ dimensions, accounting for the color channels) rather than the full image. We evaluate models using test-set bits-per-dimension (bpd); as this is normalized for dimension, our bpds are directly comparable with bpds on the entire image.

Our Monarch-HCLTs replace the dense sum blocks with Monarch matrices. In particular, we use a composition of two Monarch-2 layers (as described in Dao et al. (2022)) as we found this to improve performance for the image datasets. Analogously with HMMs for language modeling, we use the number of floating point operations per pixel as a measure of model efficiency. This is $h^2$ for a HCLT and $3h^{3/2}$ for a Monarch-HCLT (the factor of 3 instead of

2 is due to the composition). We show the scaling plot for both PC variants on ImageNet64 (YCoCg-R) in Figure 7. We find that dense PCs plateau quickly in terms of their performance, with the more compute and memory-efficient Monarch layers often having similar performance to the corresponding dense layers with the same number of hidden states. On the other hand, scaling up the number of hidden states further via Monarch parameterizations remains effective; this is because we can use a much larger hidden size with the same compute/memory.

We further show a comparison between our models and the state-of-the-art PCs on these datasets. For reference, we also show some neural baselines, namely, RealNVP (Dinh et al., 2017), Glow (Kingma & Dhariwal, 2018), and VDM (Kingma et al., 2021). It can be seen that Monarch-HCLTs show improved scaling beyond what is achievable with standard HCLTs. Our largest model, with $m = 16384$ hidden states, achieves state-of-the-art performance for PCs (Table 5), beating even LVD-PG (Liu et al., 2023b), which is an image-specialized PC-based model that uses a high-level PC over $4 \times 4$ patch PCs, together with a complex optimization procedure involving latent variable distillation (Liu et al., 2023a) for initializing the parameters and a progressive growing technique. In contrast, we simply use random initializations, the generic HCLT variable decomposition, and train end-to-end using only the well-established EM algorithm for PCs.

Despite the improvements achieved by employing Monarch matrices to scale up the hidden state size, the modeling performance still currently lags behind neural image models. Relative to language modeling, this might be because of the lack of an effective homogeneous architecture like HMMs, which limits scaling up the hidden size further. In particular QPCs, which are based on the quad-graph architecture (Mari et al., 2023), have significantly worse performance compared to HCLTs.

## 6. Related Work

Our work is connected to recent efforts in the probabilistic circuits community to find more efficient parameterizations of tensorized circuits. While early implementations of tensorized circuits used Kronecker product blocks (Peharz et al., 2020), the recent trend has been to prefer Hadamard product blocks. Loconte et al. (2024a) noted that the composition of Kronecker/Hadamard product blocks with sum blocks can be interpreted as Tucker (Tucker, 1964) / canonical-polyadic (CP) (Carroll & Chang, 1970) tensor decompositions respectively. Our work tackles an orthogonal aspect in that it focuses on the sum-to-product connection, which has thus far always been implemented as a dense matrix.

Circuit multiplication is a fundamental operation which has been studied extensively in the probabilistic circuits literature, with work on deriving theoretical conditions for tractability (Shen et al., 2016; Vergari et al., 2021; Wang et al., 2024; Zhang et al., 2025), use as a building block in various applications of PCs (Choi et al., 2015; Khosravi et al., 2019; Wang & Kwiatkowska, 2023; Zhang et al., 2024), and improving the expressivity of PCs (Loconte et al., 2024b; 2025; Wang & Van den Broeck, 2025). Our work shows how to explicitly materialize and implement these products efficiently for tensorized circuits.

There is extensive research on structured matrices for improving neural network efficiency and scalability. Qiu et al. (2024) study scaling laws across structures and introduce Block Tensor-Train (BTT), which generalizes tensor-train (TT) (Oseledets, 2011) and Monarch matrices, achieving competitive performance with lower computational costs. Potapczynski et al. (2024) extend this to BTT-MoE, a Mixture-of-Experts model that sparsifies BTT computation. While our generalized Monarch matrices align with the TT-to-BTT generalization (rank-1 case), our approach provides practical probabilistic semantics, enabling PCs with Monarch matrices to be formed by multiplying smaller PCs with dense matrices. Additionally, Sehanobish et al. (2024) propose structured unrestricted-rank matrices for efficient Transformer fine-tuning.

## 7. Conclusion

We propose scaling up probabilistic circuits by replacing dense sum blocks with structured sparse matrices. Our approach not only provides theoretical insight by establishing the connection between circuit multiplications and structured matrices but also demonstrates significant empirical improvements, including in modeling performance normalized for compute. Future work could examine sparsifying the hidden state representation to overcome the memory bottleneck, as well as utilizing the improved scaling to unlock improvements in downstream applications of PCs such as controllable language generation.

## Acknowledgements

We thank Anji Liu for assistance with the PyJuice PC library and discussions on image dataset evaluation. The Authors acknowledge the National Artificial Intelligence Research Resource (NAIRR) Pilot and TAMU ACES for contributing to this research result. This work was funded in part by the DARPA ANSR, CODORD, and SAFRON programs under awards FA8750-23-2-0004, HR00112590089, and HR00112530141, NSF grant IIS1943641, and gifts from Adobe Research, Cisco Research, and Amazon. Approved for public release; distribution is unlimited.

## Impact Statement

This paper presents work whose goal is to advance the field of Machine Learning. There are many potential societal consequences of our work, none which we feel must be specifically highlighted here.

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

# A. Interpolating Butterfly and Monarch Matrices

Butterfly matrices (Dao et al., 2019; Meng et al., 2022) are a class of expressive structured matrices. They are constructed as the product of sparse matrices known as butterfly factor matrices (Parker, 1995). In this section, we will show how Butterfly matrices (products of sparse Butterfly factor matrices) can be viewed as generalized Monarch matrices (products of dense high-dimensional tensors), as stated in Theorem 3.4. For ease of exposition, we will describe here Butterfly factor matrices of size $D \times D$, where $D$ is a power of 2.

**Definition A.1** (Butterfly Factor). Given any $D = 2^d$ and $1 \leq i \leq d$, a butterfly factor matrix $B(i, D)$ is a sparse matrix with the following sparsity pattern:

- $B(1, D)$ has non-zero elements only along the diagonals of the four $\frac{D}{2} \times \frac{D}{2}$ submatrices.

- $B(i, D)$ is a block-diagonal matrix with block size $\frac{D}{2^{i-1}} \times \frac{D}{2^{i-1}}$, where each block is a $B\left(1, \frac{D}{2^{i-1}}\right)$ butterfly factor.

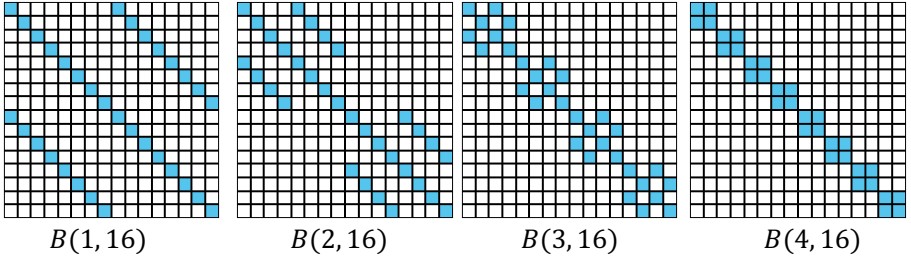

$$B(1,16) \qquad B(2,16) \qquad B(3,16) \qquad B(4,16)$$

Figure 8: Illustration of Butterfly factors (Meng et al., 2022).

Examples of butterfly factors for $D = 16$ are shown in Figure 8. Notice that each butterfly factor $B(i, D)$ for $i \in [d]$ has $2D$ nonzero elements, but different sparsity patterns. We now present a new interpretation of butterfly factors as a *reshaping* of a *dense* $(d+1)$-dimensional tensor of shape $2 \times 2 \times \cdots \times 2$. In particular, consider the following reshaping:

**Definition A.2** (Butterfly Unfurling). Let $A_{k_1, \ldots, k_{d+1}}$ be a $(d+1)$-dimensional $2 \times 2 \times \ldots \times 2$ tensor. We define the $i^{\text{th}}$ *butterfly unfurling* of $A$ to be the $2^d \times 2^d$ matrix defined by:

$$B_{j_1 j_2 \ldots j_d, j'_1 j'_2 \ldots j'_d} := A_{j_i, j_1, \ldots, j_{i-1}, j_{i+1}, \ldots, j_d, j'_i} \prod_{c \neq i} \delta^{j'_c}_{j_c} \tag{3}$$

where we interpret e.g. $j_1 j_2 \ldots j_d$ as the integer in $\{0, \ldots, 2^{d-1}\}$ represented by the binary string, and $\delta^{j'_c}_{j_c}$ is equal to 1 if $j_c = j'_c$ and 0 otherwise.

Intuitively, this takes the first and last dimensions of $A$ and inserts them into the $i^{\text{th}}$ bit of the row and column of $B$ respectively; while all other bits of $B$ between the rows and columns are tied. This produces the sparse structure of a Butterfly matrix. For instance, the $d^{\text{th}}$ butterfly unfurling creates a block-diagonal matrix with block size 2.

**Proposition A.3.** *The $i^{\text{th}}$ butterfly unfurling of a $(d+1)$-dimensional tensor is a butterfly factor matrix $B(i, 2^d)$.*

*Proof.* We begin by considering the $1^{\text{st}}$ butterfly unfurling, which is:

$$B_{j_1 j_2 \ldots j_d, j'_1 j'_2 \ldots j'_d} := A_{j_1, j_2, \ldots, j_d, j'_1} \prod_{c \neq 1} \delta^{j'_c}_{j_c}$$

Notice that the $j_1, j'_1$ are the most significant bits of the row/column respectively, and as each value of $(j_1, j'_1)$ corresponds to one of the $D/2 \times D/2$ shaped submatrices. The entry $B_{j_1 j_2 \ldots j_d, j'_1 j'_2 \ldots j'_d}$ is 0 unless $j_c = j'_c$ for all $c \neq 1$, i.e. if the entry lies along the diagonal of the submatrix.

Now let us consider the $i^{\text{th}}$ butterfly unfurling for $i \neq 1$:

$$B_{j_1 j_2 \ldots j_d, j'_1 j'_2 \ldots j'_d} := A_{j_i, j_1, \ldots, j_{i-1}, j_{i+1}, \ldots, j_d, j'_i} \prod_{c \neq i} \delta^{j'_c}_{j_c}$$

Notice that, as the most significant bits, $j_1, ..., j_{i-1}$ and $j'_1, ..., j'_{i-1}$ index the submatrices of size $\frac{D}{2^{i-1}}$. Note that the entry $B_{j_1 j_2 ... j_d, j'_1 j'_2 ... j'_d}$ is zero unless $j_c = j'_c$ for all $c = 1, .., i-1$; thus only the block submatrices along the diagonal can have nonzero elements. Now let us consider any such submatrix; w.l.o.g. let us choose $j_1 = ... = j_{i-1} = j'_1 = ... = j'_{i-1} = 0$. Then we have:

$$B_{00...0j_i...j_d, 00...0j'_i...j'_d} := A_{j_i, 00..0j_{i+1},...,j_d,j'_i} \prod_{c>i} \delta^{j'_c}_{j_c}$$

This submatrix satisfies the sparsity pattern of the $1^{st}$ butterfly unfolding of a $d - i + 1$ dimensional tensor, i.e. is a butterfly factor matrix $B(1, \frac{D}{2^{i-1}})$ as required. $\qquad\square$

Notice that, since a butterfly unfurling is just a reshaping, given any butterfly factor $B(i, D)$ we can invert the unfurling operation and get a dense tensor $A(i, D)$.

**Butterfly matrices** Butterfly matrices are constructed as the product of butterfly factors of a given dimension, i.e. $B(2^d) := B(1, 2^d)B(2, 2^d) ... B(d, 2^d)$. If we instead parameterize these matrices using a butterfly unfurling, we get a product of dense tensors with some repeated indices. For example, when $d = 2$, we get the following expression:

$$\begin{aligned}
B(4)_{j_1 j_2, j_5 j_6} &= \sum_{j_3, j_4} B(1, 4)_{j_1 j_2, j_3 j_4} B(2, 4)_{j_3 j_4, j_5 j_6} \\
&= \sum_{j_3, j_4} A(1, 4)_{j_1, j_2, j_3} \delta^{j_4}_{j_2} A(2, 4)_{j_4, j_3, j_6} \delta^{j_5}_{j_3} \\
&= A(1, 4)_{j_1, j_2, j_3} A(2, 4)_{j_2, j_5, j_6}
\end{aligned}$$

We can generalize this idea to higher dimensions:

**Theorem A.4.** *Any butterfly matrix $B(2^d)$ matrix of size $2^d \times 2^d$ can be written as:*

$$\begin{aligned}
&B(2^d)_{j_1...,j_d,j_{d+1}...j_{2d}} \\
&= A(1, 2^d)_{j_1,...,j_{d+1}} A(2, 2^d)_{j_2,...,j_{d+1}} ... A(d, 2^d)_{j_d,...,j_{2d}}
\end{aligned}$$

*where each $A(i, 2^d)$ is a $2 \times 2 \times ... \times 2$ dense tensor of dimension $d + 1$.*

*Proof.* Essentially this is replacing each butterfly factor with its furled version, and by inspection of the unfurling formula in Equation 3. For each of the butterfly factors, let us write:

$$B(i, 2^d)_{j_1^{(i)} j_2^{(i)} ... j_d^{(i)}, j_1^{(i+1)} j_2^{(i+1)} ... j_d^{(i+1)}} := A(i, 2^d)_{j_i^{(i)}, j_1^{(i)},...,j_{i-1}^{(i)}, j_{i+1}^{(i)},...,j_d^{(i)}, j_i^{(i+1)}} \prod_{c \neq i} \delta^{j_c^{(i+1)}}_{j_c^{(i)}}$$

where we have attached a superscript to each index to indicate it is associated with the $i^{\text{th}}$ butterfly factor; in particular, $j_c^{(1)} = j_c$ and $j_c^{(d)} = j_{d+c}$.

Notice that the delta functions require (for a non-zero entry) that $j_c^{(i)} = j_c^{(i+1)}$ whenever $c \neq i$. By iterative application of this rule, one can see that the indices of $A(i, 2^d)$ correspond to $j_i, ..., j_{i+d}$ (up to some permutation). For example, the first index $j_i^{(i)} = j_i^{(i-1)} = ... = j_i^{(1)} = j_i$. In general, $j_c^{(i)} = j_{d+c}$ whenever $c < i$, and $j_c^{(i)} = j_c$ whenever $c \geq i$. $\qquad\square$

Interpreting $x_{j_3 j_6}$ as a matrix $x_{j_3, j_6}$, we get a sequence of tensor contractions matching the structure of the generalized Monarch matrices. By generalizing this idea to larger $d$, we can encode and execute any butterfly matrix-vector multiplication as a series of dense tensor contractions.

# B. Additional Experimental Results and Details

## B.1. Additional experiment setup for text8

We train Monarch-HMM using two-layer Monarch matrices and a $2^{19}$ hidden state for 20 epochs. Following prior works (Zhang et al., 2024), optimization is performed using the stochastic EM algorithm with a mini-batch of 4096 and a linearly decaying learning rate from 1 to 0.

## B.2. Additional results for text8

| Hidden Size | Monarch-2 | | Monarch-3 | | HMM | | Monarch-4 | |
|---|---|---|---|---|---|---|---|---|
| | FLOPs | BPC | FLOPs | BPC | FLOPs | BPC | FLOPs | BPC |
| $2^{12} = 4096$ | 524288 | 2.041 | 196608 | 2.135 | 16777216 | 1.908 | 131072 | 2.188 |
| $2^{13} = 8192$ | 1572864 | 1.926 | 524288 | 2.013 | 67108864 | 1.786 | 327680 | 2.065 |
| $2^{14} = 16384$ | 4194304 | 1.824 | 1310720 | 1.895 | 268435456 | 1.726 | 786432 | 1.948 |
| $2^{15} = 32768$ | 12582912 | 1.747 | 3145728 | 1.814 | 1073741824 | 1.687 | 1835008 | 1.853 |
| $2^{16} = 65536$ | 33554432 | 1.690 | 8388608 | 1.741 | | | | |
| $2^{17} = 131072$ | 100663296 | 1.637 | | | | | | |
| $2^{18} = 262144$ | 268435456 | 1.609 | | | | | | |
| $2^{19} = 524288$ | 805306368 | 1.578 | | | | | | |

Table 6: We report FLOPs and test set BPC of HMM and variying Monarch structures (Monarch-2, Monarch-3, and Monarch-4) for hidden size ranging from $2^{12}$ to $2^{19}$ on dataset text8.

## B.3. Additional experimental setup for image modeling

We use hidden Chow-Liu trees (HCLT) (Liu & Van den Broeck, 2021) to define the variable decomposition (*vtree*) of the PC. We train all models using stochastic EM. In particular, we use a cosine learning rate decay scheduler over the course of the entire training. The mini-batch size $M = 20000$ and number of epochs $E = 20$ used in all experiments were chosen based on an initial hyperparameter search in the range $M \in [1000, 5000, 20000, 60000]$ and $E \in [5, 10, 20]$.

**Color Transforms**    The original ImageNet data has three color channels $R$, $G$, $B$, each taking integer values in $[0, 255]$. We apply a **lossless** YCoCg-R transform (Malvar & Sullivan, 2003) to this data to obtain three transformed color channels $Y, Co, Cg$, given as follows in integer arithmetic:

$$Co = R - B$$
$$tmp = B + \lfloor Co/2 \rfloor$$
$$Cg = G - tmp$$
$$Y = tmp + \lfloor Cg/2 \rfloor$$

where $Y$ and $Cg$ take integer values in $[0, 255]$ and $Co$ takes integer values in $[-255, 255]$. This transform is exactly invertible (lossless) and as such likelihoods on this transformed data are comparable to likelihoods on the RGB dataset.

To compare against prior PC results, we also report results using a **lossy** YCoCg transform defined as follows. Firstly, the data is normalized to take values in the range $[0, 1]$:

$$r = R/255; \ g = G/255; \ b = B/255$$

Then, the following linear transformation is applied:

$$co = r - b$$
$$tmp = b + co/2$$
$$cg = g - tmp$$
$$y = tmp * 2 + cg + 1$$

The variables $y, co, cg$ now take values in $[-1, 1]$. To convert back to categorical data, we quantize the interval $[-1, 1]$ uniformly into 256 bins and convert $y, co, cg$ into their quantized versions $Y, Co, Cg$ which take integer values in $[0, 255]$. This transformation from $R, G, B$ to $Y, Co, Cg$ is not invertible and so likelihoods on this dataset cannot be compared to the original RGB dataset.

**Image Patches**    We train PCs with 192 variables, modeling $8 \times 8$ aligned image patches (with 3 color channels). This data is obtained by splitting each image from the the ImageNet32 (resp. ImageNet64) dataset into 16 (resp. 64) of these patches.

