# OpenReview forum: "Scaling Probabilistic Circuits via Monarch Matrices"
_ICML.cc/2025/Conference — ICML 2025 poster_

### Official Review · Reviewer_zQvJ · 2025-02-15

**Overall Recommendation:** 2

**Summary:**

This paper replaces dense matrices with sparse Monarch matrices, reducing the computation cost and maintaining accuracy.

**Claims And Evidence:**

Yes, the claims made in the submission are supported by clear and convincing evidence.

**Essential References Not Discussed:**

No.

**Experimental Designs Or Analyses:**

I have checked the soundness/validity of any experimental designs or analyses.

**Methods And Evaluation Criteria:**

The proposed methods may be suitable for the problem or application at hand.

**Other Comments Or Suggestions:**

1. Enhancing Tables 4 and 5 with additional results would further improve the completeness of the paper.
2. Also, could you show me the model size in the Table?

**Other Strengths And Weaknesses:**

Strengths:
1. The paper is exceptionally well-written and easy to follow.
2. This paper presents some theoretical analysis.
3. This paper replaces dense matrices with sparse Monarch matrices to reduce memory and computation cost.

Weaknesses:
1. To be honest, I do not know much about hybrid models since I have only read several papers such as Mamba. However, I think the main issue of the hybrid model is that we cannot scale up the size of the hybrid model. That is my understanding. Consequently, almost no leading companies choose to train and deploy hybrid models.
2. As for the paper, could you show me more results if you increase the length of the example from 256 to 1024?
3. In addition, could you show me the results evaluated over imagenet-256?

**Questions For Authors:**

See strengths and weaknesses.

**Relation To Broader Scientific Literature:**

This paper presents several interesting results.

**Theoretical Claims:**

Yes, I have checked the proofs for correctness. Any minor issues present should not impede understanding of the overall paper.

---

> ### Author Rebuttal · Authors · 2025-04-01
>
> Thank you for your feedback.
>
> ```
> To be honest, I do not know much about hybrid models since I have only read several papers such as Mamba. However, I think the main issue of the hybrid model is that we cannot scale up the size of the hybrid model. That is my understanding. Consequently, almost no leading companies choose to train and deploy hybrid models.
> ```
>
> To clarify, in this work we investigate scaling of probabilistic circuit models. As opposed to hybrid models (and indeed transformers), probabilistic circuits have the key property of tractability, which enables efficient computation of quantities relating to the probability distribution such as marginals. Tractability has been used to achieve state-of-the-art results in applications such as controllable generation [1] and image inpainting [2], among others, beating transformer and diffusion-based models. Thus, we respectfully believe that research on alternative directions for generative modeling is valuable and this should not be viewed as a weakness of the work.
>
>
> ```
> As for the paper, could you show me more results if you increase the length of the example from 256 to 1024?
> ```
>
> We appreciate the reviewer’s request for further experiments, but we do not have the computational resources to run the requested experiments during the rebuttal period. Please note that we chose the sequence lengths and ImageNet downscaled resolutions to match with those considered by our baselines for more direct comparison; in particular, our method shows significant and consistent advantage over the PC baselines. Given the existing range of experiments we do not believe that there would be any change in the qualitative conclusions.
>
>
> ```
> Also, could you show me the model size in the Table?
> ```
>
> The PCs for image modeling have around 1.5B parameters. The Monarch HMM on text8 with hidden size $h=2^{19}$ has 0.75B parameters and the Monarch HMM on lm1b has 4.75B parameters. We will update the Tables with model sizes.
>
>
> [1] Zhang et al. “Adaptable Logical Control for Large Language Models” NeurIPS 2024
>
> [2] Liu et al. “Image Inpainting via Tractable Steering of Diffusion Models” ICLR 2024

---

> > ### Comment · Reviewer_zQvJ · 2025-04-02
> >
> > Thanks for your response! As I mentioned earlier, I am not very familiar with hybrid models. I am eager to hear the insights and suggestions from **Reviewer AzZ2** and **Reviewer Gwtt**.

---

### Official Review · Reviewer_AzZ2 · 2025-03-02

**Overall Recommendation:** 4

**Summary:**

This paper proposes a novel parameterization for probabilistic circuits (PCs) to improve their scalability, using structured sparse matrices called Monarch matrices. By replacing dense matrices in sum blocks of PCs with Monarch matrices, the proposed methods can reduce computational costs and allow larger scale PC training. The authors conducted experiments on text and image datasets. To my knowledge, the proposed idea is novel and the experimental results look compelling (at least compared to other PC methods).

**Claims And Evidence:**

The claims made in the paper are well supported by the conducted experiments. The authors are honest about the remaining gap between the proposed method and state-of-the-art generative modeling methods.

**Essential References Not Discussed:**

To my knowledge, no.

**Experimental Designs Or Analyses:**

The experiments are reasonably designed and analyzed for the studied problem of the paper.

**Methods And Evaluation Criteria:**

The proposed methods and evaluation criteria (efficiency in terms of FLOPs and performance in terms of test log-likelihood) are reasonable to the problem studied in this paper.

**Other Comments Or Suggestions:**

N/A.

**Other Strengths And Weaknesses:**

Strengths:
- This paper advances scalable PC training and shows empirically compelling results, which is critical to the PC community.

Weaknesses:
- The memory saveup shown in Table 3 is not significant, which limits the extension of the method.
- There is still a large performance gap between the proposed method and other SOTA methods (e.g. diffusion models). In image modeling experiments, the method is weaker than flow-based models which have been proposed years ago.
- I don't think the sampling time comparison throughout the paper is convincing. According to the paper, the baseline results are often taken from the original papers. The comparison in such case is not fair under varied implementation details and hardware setup.

**Questions For Authors:**

- I find the introduction of Butterfly matrices in line 205 a bit redundant and irrelevant. Why are they mentioned in the methods section rather than the related work? Have they been applied in PCs?
- Why is the YCoCg transform used in the image modeling experiments? Is it necessary in the setup of this paper? If not, how does the proposed method perform using the original RGB images?

**Relation To Broader Scientific Literature:**

The key contributions of the paper build upon [1] and the implementations and training of the proposed method rely on [2] and [3].


[1] Dao et al. Monarch: Expressive structured matrices for efficient and accurate training. ICML 2022.

[2] Peharz et al. Einsum Networks: Fast and Scalable Learning of Tractable Probabilistic Circuits. ICML 2020.

[3] Liu et al. Tractable regularization of probabilistic circuits. NeurIPS 2021.

**Theoretical Claims:**

I checked the derivations for equations (1) and (2) and they look correct to me. But the flow from the Monarch matrix operations to the construction of a PC is not clear to me by reading sections 3 and 4.

---

> ### Author Rebuttal · Authors · 2025-04-01
>
> Thank you for your insightful feedback.
>
> ```
> I find the introduction of Butterfly matrices in line 205 a bit redundant and irrelevant. Why are they mentioned in the methods section rather than the related work? Have they been applied in PCs?
> ```
>
> One of the main contributions of this work is identifying the connection between structured Monarch matrices and the multiplication of PCs: the product of two dense PCs gives exactly the Monarch matrix while the product of $d$ dense PCs of hidden size 2 gives the Butterfly matrix.
>
> Butterfly matrices are a well-known type of structured matrix which require only $O(h\log h)$ compute/memory (compared with $O(h^{3/2})$ for Monarch); however, despite their theoretical efficiency, Butterfly matrices are known to be less efficient on modern GPUs (due to their sparsity).
>
> Our construction of Monarch matrices as circuit product immediately implies that (i) one can interpolate between Butterfly and Monarch matrices through the generalized Monarch matrix; and (ii) this corresponds precisely to the multiplication of k PCs, for integers k >= 2. This then motivated us to perform an empirical study for $k = 2, 3, 4$ to see if one could achieve better scaling using the generalized Monarch matrix.
>
> We will incorporate this explanation to motivate this section more clearly.
>
> ```
> The memory saveup shown in Table 3 is not significant, which limits the extension of the method.
> ```
>
> We would like to note that Table 3 shows the **training** memory consumption with a reasonably large batch size $B = 128$ and the # of variables $n = 256$. At **inference** time (i.e. no gradient computation), Monarch PCs can be significantly more memory-efficient: with $h = 2^{18}$, a dense PC would still use $\approx 256$ GB while a Monarch PC (2-layer) would only use $\approx 1$ GB. That is, even when training a large Monarch PC is relatively memory-consuming, at inference time, researchers can still apply large powerful Monarch PCs with very little cost.
>
> In addition, the seemingly inefficient memory consumption of Monarch PCs during training is due to the need of gradient computation, which can be optimized by various well-known techniques such as gradient checkpointing, mixed-precision training, etc. Further, as shown at the end of Sec. 5.2, the hidden states of large Monarch PCs are actually sparse, motivating future research to exploit such sparsity to reduce the memory consumption of training large Monarch PCs.
>
> ```
> There is still a large performance gap between the proposed method and other SOTA methods (e.g. diffusion models).
> ```
>
> We agree that there is still a large gap between state-of-the-art PCs and the other deep generative models, but at the same time we would like to highlight that we have already scaled PCs to a degree far beyond existing tractable probabilistic models, while the SotA generative models are substantially less tractable compared to PCs (e.g. none of transformers, normalizing flows or diffusion models allow for tractable computation of marginal probabilities). Closing the performance gap between PCs and other deep generative models is precisely the motivation of our research.
>
> ```
> I don't think the sampling time comparison throughout the paper is convincing.
> ```
> The sampling time comparison in Table 4 is not meant to argue that PCs are superior to discrete diffusions in terms of efficiency but more of a sanity check: for readers who are not particularly familiar with PCs, we just want to show that our models can be efficiently implemented. More specifically, for example, the D3PM Uniform model achieves $\leq 1.61$ bpc with 1000 diffusion steps (3.6s) and $\leq 1.79$ bpc with 20 diffusion steps (0.077s), where the latter number is more comparable to that of Monarch HMMs. In our revision, we will add both D3PM results to the table and carefully rephrase our statement about the runtime of our models. Thank you for your suggestion.
>
> ```
> Why is the YCoCg transform used in the image modeling experiments?
> ```
> Table 5 shows the experiment results on both images transformed via YCoCg and transformed via YCoCg-R (the Lossless column). Since the YCoCg-R transform is lossless, the BPDs are directly comparable to the original RGB dataset. The two strongest PC baselines [3, 4] only reported their results on YCoCg transformed images (which is not reported in those papers, but is confirmed in [5] and the software implementations); we choose to report results for both settings for a fair comparison to [3, 4] and to models trained on RGB datasets, following the practice of [5].
>
> [3] Liu et al. "Scaling Up Probabilistic Circuits by Latent Variable Distillation." ICLR 2023
>
> [4] Liu et al. "Understanding the distillation process from deep generative models to tractable probabilistic circuits." ICML 2023
>
> [5] Gala et al. "Scaling Continuous Latent Variable Models as Probabilistic Integral Circuits." NeurIPS 2024

---

### Official Review · Reviewer_NJT4 · 2025-03-14

**Overall Recommendation:** 4

**Summary:**

Despite many advantages of probabilistic circuits (PC), their implementations are often difficult due to computational burden, even with block structures. In this paper, the authors proposed an alternative method that replaces dense sum blocks with Monarch matrices, and the method significantly reduce the memory and computation costs. Ultimately, the authors claimed that this method  significantly bridges the gap between highly tractable models (probabilistic circuits) and less tractable models (diffusion models).

**Claims And Evidence:**

Claim 1. Monarch matrices are efficient.
This claim is well-supported via theoretical arguments on Page 3 by showing the improvement from $O(m^4)$ to $O(m^3)$ edges.

Claim 2. The Monarch models have outstanding performance.
The performance is supported via empirical tests, shown in Tables 1 and 5.

Claim 3. The Monarch models have better scaling behavior.
The performance is supported via empirical tests, shown in Figures 4 and 7.

**Essential References Not Discussed:**

I am not qualified to comment on this section.

**Experimental Designs Or Analyses:**

The results (in tables and figures) support the paper's claims, but I am not qualified to comment the empirical part.

**Methods And Evaluation Criteria:**

This paper has extensive justifications in theory. I am not knowledgable in experiments, but the benchmark datasets are standard for generative models. Using FLOPs to measure efficiency is a commonly accepted choice.

**Other Comments Or Suggestions:**

I do not have any other comments or suggestions.

**Other Strengths And Weaknesses:**

Strengths:
1. The paper is well-motivated, studies an important problem, and has rigorous theoretical justifications.
2. The claim on bridging performance gaps is justified and promising to future research in this direction.
3. The empirical comparisons take care of a wide range of metrics.

**Questions For Authors:**

I do not have any other questions.

**Relation To Broader Scientific Literature:**

This paper extensively discusses its connections with relevant fields, including previous works on probabilistic circuits (in particular block parameterizations) and relevant background in linear algebra (butterfly matrices and Monarch matrices). They also discussed other types of models such as the diffusion model and flow-based models, and gave a high level picture on the status of PCs.

**Theoretical Claims:**

I have checked all definitions and linear algebraic results: Discussions on page 3, and the theorems and proofs in Appendix A.

---

> ### Author Rebuttal · Authors · 2025-04-01
>
> Thank you for your encouraging feedback. Please feel free to follow up if you have any questions.

---

### Official Review · Reviewer_Gwtt · 2025-03-14

**Overall Recommendation:** 2

**Summary:**

This paper introduces a novel method to scaling Probabilistic Circuits (PCs) by replacing dense matrices in sum nodes with Monarch matrices which is a type of structured sparse matrices constructed by Kronecker products. The key idea is to leverage the sparsity and structure of Monarch matrices to reduce memory and computation costs which is demonstrated on generative modeling benchmarks. Furthermore, this paper provides a theoretical foundation on the connection between Monarch matrices and circuit multiplication.

**Claims And Evidence:**

The majority of the claims in the paper are supported by clear evidence, such as those related to computational efficiency, performance improvements.

**Essential References Not Discussed:**

I think this paper has discussed enough relevant work in this area and provided comprehensive references to previous studies to illustrate its main contributions.

**Experimental Designs Or Analyses:**

Yes, the experiments support the claims made in the paper, but additional analyses could further strengthen the results.

**Methods And Evaluation Criteria:**

The methods and evaluation criteria make sense for the problem and application at hand, but evaluating on larger datasets and broader tasks could further strengthen the claims.

**Other Comments Or Suggestions:**

I believe the first two weaknesses highlight the main issues. As for the third weakness, since it involves additional experiments, if there isn’t enough time to conduct them, a detailed verbal explanation of the method’s generalization would be helpful.

**Other Strengths And Weaknesses:**

strength:
1. The paper introduces Monarch matrices as a structured sparse parameterization for sum blocks in PCs, bridging the gap between tractability and scalability.
2. The paper provides solid theoretical justification for using Monarch matrices, linking them to circuit multiplication and structured sparsity.
3. Comprehensive experiments on generative modeling benchmarks (Text8, LM1B, ImageNet) demonstrate state-of-the-art performance with reduced computational cost (FLOPs).

weakness:
1. The paper does not compare Monarch matrices to alternative structured representations like Block Tensor-Train (BTT) decomposition [1] or Toeplitz-like structured layers. How do Monarch matrices perform relative to these alternatives in terms of efficiency and expressiveness?
2. The paper highlights performance improvements but does not discuss potential failure cases. For example, does the structured sparsity of Monarch layers introduce expressiveness limitations compared to dense PCs?
3. The experiments focus on generative modeling benchmarks, but PCs are also used in other applications, such as causal inference, fairness, and tractable reasoning. Evaluating Monarch-based PCs in non-generative tasks (e.g., probabilistic reasoning) would further strengthen the paper's impact.

[1] Qiu et al. Compute better spent: Replacing dense layers with structured matrices. ICML 2024.

**Questions For Authors:**

Please check my questions in the weakness section.

==Post Rebuttal==

I think the empirical evaluation of this paper can be more sufficient. I am ok if AC and the other reviewers decide to accept this paper.

**Relation To Broader Scientific Literature:**

This paper builds upon and extends multiple streams of prior research in PCs, structured matrices, and efficient model scaling.

**Theoretical Claims:**

The theoretical framework is sound, but further formalization would strengthen the paper's contributions.

---

> ### Author Rebuttal · Authors · 2025-04-01
>
> Thank you for your feedback.
>
> ```
> The paper does not compare Monarch matrices to alternative structured representations like Block Tensor-Train (BTT) decomposition [1] or Toeplitz-like structured layers.
> ```
>
> Our study is not only limited to the Monarch matrices defined in [1]. Our construction of Monarch matrices as product of PCs with dense layers naturally generalizes the definition of Monarch matrices: the construction in [2] corresponds to the special case of multiplying **2** dense PCs, and we consider Monarch structures obtained by multiplying **$k$** dense PCs, showing empirical results for k = 2, 3, 4.
>
> Secondly, in terms of BTTs, we would like to note that BTTs of rank 1 correspond exactly to the Monarch matrices proposed by [2], and [1] made an important observation that BTTs of rank higher than 2 do not lead to better scaling behaviors while increasing higher memory cost.
>
> As a quick corroboration for PCs, we studied PCs with BTT layers on the ImageNet32 (lossless) dataset. As shown in the Figure at https://anonymous.4open.science/r/MonarchRebuttal-D0D3/a.pdf, we compare Monarch PCs with BTT layers of rank 2, 4, and 8: BTT-2 has a similar scaling curve with Monarch, and the scaling curves of BTT-4 and BTT-8 get worse as rank increases, echoing the findings of [1].
>
> Finally, we would like to note that prior work on PCs has not considered using non-dense matrices, and our work opens up a new way of scaling PCs with great promise as illustrated by empirical achievements. We believe that investigating Toeplitz-like structured layers or other efficient representations would be excellent topics for future work.
>
> ```
> The paper highlights performance improvements but does not discuss potential failure cases.
> ```
>
> We agree that if we fix the hidden size of PCs, replacing the dense linear layers with Monarch matrices indeed reduces the expressive power, which is exactly shown in Figure 5, where we measure the performance of PCs with varying hidden sizes and different structures of linear layers. We can see that when fixing the hidden sizes, dense matrices always perform better, but at the same time, as the hidden size grows: (1) the performance gap between the dense PCs and the Monarch PCs diminishes rapidly while (2) the FLOPs gap between dense PCs ($O(h^2)$) and Monarch PCs($O(2h^{3/2}$) grows rapidly; and these two factors together lead to the large gap between the scaling curves of the dense PCs and Monarch PCs shown in Figure 4.
>
> From a theory perspective, one could ask whether there exists any distribution represented as compact dense PCs which would require exponential size to represent as a Monarch PC. The answer is no, as we have shown that a Monarch circuit can be interpreted as a relaxed version of the product of two circuits. Thus, any dense PC can be simulated by multiplying it with a PC representing the uniform distribution.
>
> We will incorporate this discussion into our revision and highlight that our study on the expressive power of Monarch matrices in PCs is focused on the empirical side.
>
> ```
> The experiments focus on generative modeling benchmarks, but PCs are also used in other applications … Monarch-based PCs in non-generative tasks (e.g., probabilistic reasoning) would further strengthen the paper's impact.
> ```
>
> Prior work has shown that the better the PCs model the desired distributions, the better they perform on downstream applications. For example, one line of work [3, 4] on applying PCs for controllable text generation from LLMs shows that the better the PCs approximate the LLMs, the higher the text generation quality, as shown in Figure 3 of [3] and Table 1 of [4]. Similarly, in the application of PCs to group fairness, Figure 3 and Table 1 of [5] show that the PC model achieving the best likelihood also leads to better classification accuracy and lower discrimination scores. Based on these findings, we believe that Monarch PCs, by achieving significantly better generative modeling performance, should naturally give rise to better downstream performance.
>
> Hence, instead of testing our state-of-the-art PCs on downstream applications, we devote our effort, as well as the limited computation resources, to ablation studies that may help people better understand the behavior of PCs with Monarch matrices, such as: what is a good number of dense PCs to multiply to form Monarch structures, how does initialization of PC parameters via circuit products benefit training, is the hidden states of PCs also sparse and etc.
>
> [1] Qiu et al. “Compute Better Spent: Replacing Dense Layers with Structured Matrices” ICML 2024
>
> [2] Dao et al. “Monarch: Expressive Structured Matrices for Efficient and Accurate Training” ICML 2022
>
> [3] Zhang et al. “Tractable Control for Autoregressive Language Generation” ICML 2023
>
> [4] Zhang et al. “Adaptable Logical Control for Large Language Models” NeurIPS 2024
>
> [5] Choi et al. “Group Fairness by Probabilistic Modeling with Latent Fair Decisions” AAAI 2021

---

### Decision · Program_Chairs · 2025-05-01

**Decision:**

Accept (poster)

**Comment:**

This paper introduces a novel parameterisation for the sum blocks in probabilistic circuits using Monarch matrices.  Monarch matrices allow for sparsity and tensorized operations, and thus their use significantly improves the scalability of probabilistic circuits.  The reviews were rather bi-modal with two accepts and two weak rejects (4, 4, 2, 2).  One of the weak reject reviews was rather short and signaled low confidence or even misunderstanding about the subject area.  Reviewers found that the paper was "exceptionally" well written, easy to follow, well-motivated, theoretically sound and empirically strong.  One reviewer would have liked to see comparisons to other parameterizations such as Toeplitz matrices or Block Tensor-Train matrices and evaluation of probabilistic circuits on other applications beyond generative modeling.  Another reviewer pointed out that while the results seem to be good for probabilistic circuits, they still lag behind other common generative models.
Overall it seems that the work is novel, well-motivated and convincingly enables better scaling of probabilistic circuits.   Therefore, the recommendation is to accept.